# Changes of Metabolites and Gene Expression under Different Feeding Systems Associated with Lipid Metabolism in Lamb Meat

**DOI:** 10.3390/foods10112612

**Published:** 2021-10-28

**Authors:** Bo Wang, Xingang Zhao, Zhen Li, Hailing Luo, Hao Zhang, Yanping Guo, Can Zhang, Qing Ma

**Affiliations:** 1State Key Laboratory of Animal Nutrition, College of Animal Science and Technology, China Agricultural University, Beijing 100193, China; wangboforehead@163.com (B.W.); 1404010216@cau.edu.cn (X.Z.); lizhen6394@126.com (Z.L.); guoyp@cau.edu.cn (Y.G.); zhang_can@cau.edu.cn (C.Z.); 2Beijing Advanced Innovation Center for Food Nutrition and Human Health, College of Food Science and Nutritional Engineering, China Agricultural University, Beijing 100083, China; zhanghaocau@cau.edu.cn; 3Research Center of Grass and Livestock, Ningxia Academy of Agriculture and Forestry Science, Yinchuan 750002, China; maqing1973@126.com

**Keywords:** lamb meat, lipid metabolism, fatty acids, metabolites, gene expression, feeding system

## Abstract

The effects of the different feeding systems, graze feeding (GSF), time-limited graze feeding (GF), and stall-feeding (SF)) on the fatty acid content, metabolites, and genes expression of the *longissimus dorsi* (*LD*) in Tan lambs were investigated in the present study. Thirty-nine 4-month-old male Tan lambs with similar body weight (24.91 ± 1.74 kg) were selected and divided into the three feeding systems (*n* = 13) randomly. Lambs were slaughtered after 83 days of the feeding trails, and *LD* muscle samples were collected for further analysis. The results indicated that different feeding systems have no significant effect on short-chain fatty acids in Tan lambs (*p* > 0.05). However, the total saturated fatty acids (∑SFA) and monounsaturated fatty acids (∑MUFA) in the GSF and GF groups were lower than those in the SF group (*p* < 0.001). The total polyunsaturated fatty acids (∑PUFA) in the GSF group were higher than those in the GF and SF groups (*p* < 0.001). Moreover, in the comparison of both GF vs GSF groups and SF vs GSF groups, metabolomic analysis showed that metabolites such as cis-(6,9,12)-linolenic acid, arachidic acid, acetylcarnitine, and L-carnitine with lower concentration were significantly enriched in the biosynthesis of unsaturated fatty acid pathway (*p* < 0.05), but metabolites such as phosphorylcholine, glycerophosphocholine, cytidine 5’-diphosphocholine, and glycerol-3-phosphate with higher concentrations were enriched in the glycerophospholipid metabolism pathway. KEGG (Kyoto Encyclopedia of Genes and Genomes) analysis of the results indicated that in the comparison of the GSF group with the SF group, differentially expressed genes (DEGs) such as *LIPC*, *ERFE*, *FABP3*, *PLA2R1*, *LDLR*, and *SLC10A6*, were enriched in the steroid biosynthesis and cholesterol metabolism pathways. In addition, differential metabolites and genes showed a significant correlation with the content of ∑SFA, ∑MUFA, and ∑PUFA in lamb meat (*p* < 0.05). These findings demonstrated that the feeding system was an important factor in regulating fatty acid content by affecting lipid-metabolism-related metabolites and gene expression in muscle, and graze-feeding system provided lamb meat with higher ∑PUFA content than time-limited-grazing and stall-feeding systems.

## 1. Introduction

Lamb meat is an important meat product, even though it has lower consumption than other meats such as beef, pork, and chicken. Its unique flavor and nutritive value are popular with consumers. However, due to the concept of a healthy diet attracting people’s attention, more consideration is paid to the meat quality, especially the fat content and composition, which shows a tight relationship with human health [1,2]. Lipid metabolism in muscle is important for fat content and composition as well as dietetic value and sensory traits of lamb meat. Researchers have reported that lipid compounds (such as phosphatidylcholine, triglycerides, saturated fatty acids, and unsaturated fatty acids) and their reactions with other ingredients are tightly related to the odor and flavor in the raw or cooked meat [3,4,5,6]. In addition, intramuscular fat, as one of the main attributes to evaluate meat quality, is correlated with a tendency to lower water-holding capacity, but better palatability, juiciness, tenderness, and flavor precursors [7,8]. Collectively, lipid metabolism is directly or indirectly involved in the regulation of meat quality and consumer preference.

Many factors affect lipid metabolism, as well as the organoleptic qualities and acceptability of lamb meat to the consumer, such as sex, age, breed, and feeding systems [9,10,11]. The feeding system is one of the main factors affecting the meat quality, having an effect on muscle growth, muscle and fat ratio, fat accumulation, and the fatty acid (FA) composition [12]. The differences of feeding systems on FA composition from grain and grass-based diets have been widely demonstrated, and results suggest that grass-fed ruminants have higher content of unsaturated fatty acids and lower n-6/n-3 ratio than grain-fed animals [13,14], which means meat from grazing animals is favorable to the present human dietary guidelines. However, a limited number of researchers have reported the metabolites and genes involved in the regulation of lipid metabolism under different feeding systems. Recently, as a high-throughput bioanalytical method, metabolomics has become an effective technology for detection of important metabolites in ruminants [15,16], which will help to explore the underlying mechanism of the different FA contents from different feeding systems.

Tan lamb is very popular among consumers because of its unique flavor and delicious meat quality. Current feeding systems of Tan lambs in China are mainly categorized as all-day grazing, time-limited grazing with supplementary feeding, and stall-feeding. Our previous study found that the content of polyunsaturated fatty acids such as C18:3n3 (α-linolenic acid), C20:5n3 (eicosapentaenoic acid, EPA), C22:6n3 (docosahexaenoic acid, DHA), C18:2n6 (linoleic acid), C20:3n6 (eicosatrienoic acid), and C20꞉4n6 (arachidonic acid) in Tan lambs from graze feeding were higher than that in lambs from stall feeding, and showed a lower n-6/n-3 ratio [17], while the underlying mechanism of these differences is still unclear. Therefore, the objective of this study was to investigate the effect of different feeding systems on FA content, metabolites, and gene expression and their relationship in Tan lambs.

## 2. Materials and Methods

The protocol used throughout the study was approved by the Institutional Animal Care and Use Committee of the China Agricultural University and was in accordance with the Animal Ethics Committee of Beijing, P.R. China.

### 2.1. Experimental Design and Sample Collection

The experiment was conducted in the desert grassland in Yanchi County, Wuzhong City, Ningxia (longitude 107° E, latitude 37° N). During the experiment period, the grassland was dominated with two grass species—*Lespedeza* sp. and *Caranana* sp. Some other forage crops such as *Salsola* sp. and *Artemisia* sp. were distributed irregularly in the meadow. A total of 39 4-month-old male Tan lambs with similar body weight (24.91 ± 1.74 kg) were selected and randomly divided into one of the three feeding systems—all-day grazing group (GSF, *n* = 13), time-limited graze-feeding group (GF, *n* = 13), and stall-feeding group (SF, *n* = 13). The experimental period was 83 days with a 10-day pre-feeding period. Final body weight in the GSF, GF, and SF groups was 31.21 ± 2.78, 34.78 ± 2.58, and 39.02 ± 3.48 kg, respectively [17]. The grazing place was the natural grassland around the Tan-lamb farm and the supplementary diet components and nutrition levels are shown in Table 1.

The conditions of different feeding systems were as follows: the GSF group grazed from 7:00 to 19:00 every day and the GF group grazed from 7:00 to 11:00 every morning, and single pen feeding is conducted at 17:00 in the afternoon with free access to supplementary diet. The supplementary feed was adjusted during the pre-feeding period and the residual feed after daily supplementary feeding for GF group was maintained as 5–15% of total feed. All lambs in the SF group were housed in individual pens and the diet was given at 8:00 and 17:00 daily. Lambs had free access to clean water throughout the experiment.

All feeding groups were taken to Ningxin Tan-sheep slaughterhouse in Yanchi County, Ningxia, and slaughtered using standard commercial procedures according to AVMA guidelines [18]. Tan lamb was fasted for 24 h and prevented from drinking for 12 h before slaughter. The carcass weight in the GSF, GF, and SF groups was 14.68 ± 1.59, 17.06 ± 1.66, and 20.74 ± 1.71 kg, respectively [17]. After slaughter, about 300 g of the *LD* sample was taken from the 8th to 12th ribs and stored in a Ziplock bag at −20 °C for the determination of FA content. Another two samples of about 5 g were collected into RNase-free tubes and stored in liquid nitrogen for metabolite and gene expression determination.

### 2.2. Analysis of Short-Chain Fatty Acid Composition

Approximately 100 mg of *LD* muscle sample (adipose tissue on the surface of *LD* muscle was removed) was thawed at room temperature and homogenized with 2 mL pre-cooled isopropyl ether. The mixture was left at room temperature for 20 min before centrifugation at 8000× *g* at 10 °C for 15 min. A total of 800 µL of supernatant was added into a 2 mL glass centrifuge tube, followed by adding 150 µL 15% phosphoric acid solution. After then, the sample was vortex mixed for 2 min and centrifuged at 12,000 rpm for 10 min at 4 °C. The supernatant (600 µL) was collected and mixed with equal amount of ethyl acetate, then vortexed for 2 min and centrifuged at 12,000 rpm for 10 min at 4 °C. The upper phase was taken and mixed with 150 µL 5 µg/mL internal standard solution (4-methylvaleric acid) for short-chain fatty acid determination by gas chromatography (HP-INNOWAX, 30 m × 0.25 mm, ID × 0.25 μm) [19]. The temperature at inlet and ion source was set as 250 °C. The oven temperature was first set at 90 °C, then increased at a rate of 10 °C/min to 120 °C, which is followed by 5 °C/min increase till 150 °C. After that, the temperature was increased at a rate of 25 °C/min to 250 °C and held for 2 min. The analysis of the of short-chain fatty acid data was conducted by LipidSearch software version 4.1 (Thermo Scientific™). The approximate content of acetic acid (C2), propionic acid (C3), isobutyric acid (C4), butyric acid (C4), isovaleric acid (C5), valeric acid (C5), and hexanoic acid (C6) was quantified by the comparison of their chromatographic peak areas with the n-alkane internal standard.

### 2.3. Analysis of Muscle Fatty Acid Composition

Briefly, intramuscular fat from approximately 200 mg *LD* muscle sample was extracted by using a chloroform/methanol mixture [20]. The fatty acid methyl esters (FAMEs) were prepared according to the method described by Morrison and Smith [21], combing the extracted fat with 0.5 N sodium hydroxide/methanol, 20% BF_3_, and internal standard (undecanoic acid; C11:0 methyl ester). The Agilent 6890 gas chromatograph (Agilent Technologies, Santa Clara, CA, USA) equipped with a DB-23 capillary column (60.0 m × 0.25 mm, ID × 0.25 μm) and a flame ionization detector was used for the FA composition determination according to the procedure of previous studies [22]. The chromatographic model was as follows: the capillary column was programmed to run at 130 °C for 1 min, then increased at a rate of 4 °C/min to 260 °C, the column injector temperature was 270 °C. The split ratio of automatic split injector was 30:1 and the detector temperature was 270 °C. The helium flow rate was 2.0 mL/min and 1 mL of C11:0 (1.0 mg/mL) was used as internal standard solution. The FAs were identified by the comparison of retention times with the known standards (Supelco 37 Component FAME Mix, Catalogue number 18919-1AMP, Sigma-Aldrich Inc., St. Louis, MO, USA). The approximate content of FA was calculated according to the chromatographic peak area. FA content was expressed as an mg/100 g of the fresh meat in each sample.

### 2.4. Metabolomics Analysis

Approximately 100 mg *LD* muscle of Tan lamb was acquired, followed by adding 10 times volume of pre-cooled 90% methanol. The mixture was frozen with liquid nitrogen, which was then homogenized and sonicated twice at low temperature for 30 min. The homogenate was left at −20 °C for 60 min, before being centrifuged at 13,000× *g* at 4 °C for 15 min. The supernatant was freeze-dried and stored at −80 °C for analysis. To dissolve the freeze-dried meat powder, 100 μL acetonitrile aqueous solution (acetonitrile: water = 1:1, *v/v*) was added, the mixture was vortexed, and centrifuged at 14,000× *g* at 4 °C for 15 min. The supernatant was injected into ultra-high-performance liquid chromatography (UHPLC) equipped with an HILIC column (Agilent 1290 Infinity LC). The column temperature was set as 25 °C, the flow rate was 0.3 mL/min, and the injection volume was 2 μL. Mobile phase A was 25 mM ammonium acetate and 25 mM ammonia aqueous solution. Mobile phase B was acetonitrile. The gradient elution program was set as follows: 0–1 min, 85% B; 1–12 min, 85% B to 65% B; 12–12.1 min, 65% B to 40% B; 12.1–15 min, 40% B; 15–15.1 min, 40% B to 85% B; and 15.1–20 min, 85% B. The sample was placed in the auto-sampler (4 °C) during the entire analysis. A random sample injection order was adopted for continuous analysis to avoid fluctuation of instrumental detection signal. QC samples were inserted into the sample queue to evaluate the stability of the system and the reliability of experimental data. The samples were separated by UHPLC and analyzed by Triple TOF 6600 mass spectrometer (AB SCIEX). Electrospray ionization (ESI) positive-ion and negative-ion modes were used for detection. The ESI source conditions after HILIC chromatographic separation were set as follows: ion source gas1 (Gas1): 60; ion source gas2 (Gas2): 60; curtain gas (CUR): 30; source temperature: 600 °C; IonSapary Voltage Floating (ISVF) ± 5500 V (both positive and negative modes); TOF MS scan m/z range: 60–1000 Da; product ion scan m/z range: 25–1000 Da; TOF MS scan accumulation time: 0.20 s/spectra; and product ion scan accumulation time 0.05 s/spectra. The secondary mass spectrum was obtained by information-dependent acquisition (IDA) with high sensitivity mode. Declustering potential (DP): ±60 V (both positive and negative modes); collision energy: 35 ± 15 eV; IDA setting: exclude isotopes within 4 Da; and candidate ions to monitor per cycle: 6.

### 2.5. Transcriptomics Analysis

The total RNA was extracted from the *LD* muscle samples by using Trizol Reagent (Tiangen Biochemical Technology Co., Ltd., Beijing, China). The concentration and purity of the extracted RNA were detected by Nanodrop2000, the integrity of the RNA was detected by agarose gel electrophoresis, and the RIN value was determined by Agilent 2100. A single library construction required total RNA ≥ 1 µg, concentration ≥ 35 ng/μL, OD 260/280 ≥ 1.8, and OD 260/230 ≥ 1.0.

Magnetic beads with oligo (dT) were applied to perform A–T base pairing with the poly-A tail of the extracted mRNA, which was then randomly fragmented by fragmentation buffer. The extracted mRNA was separated into small fragments (about 300 bp) through magnetic bead screening. Six-base random primers (random hexamers) were added to reversely synthesize one-strand cDNA using mRNA as a template, which was followed by two-strand synthesis to form a stable double-strand structure. The double-stranded cDNA structure had a sticky end, which was converted to a blunt end by adding End Repair Mix. An “A” base was added to the 3’ end of the cDNA to connect it to a Y-shaped linker. The cDNA was amplified for 15 cycles through PCR and 2% agarose gel was used to recover the target band, which was quantified by TBS380 (Picogreen, Thermo Fisher Scientific, Shanghai, China). Clusters were generated by cBot bridge PCR amplification, which was then sequenced by Illumina Novaseq 6000 platform.

The generated raw reads were transformed to clean reads after quality control processing. Then the clean reads were mapped to the reference genome of *Ovis aries* (Oar_v4.0, https://www.ncbi.nlm.nih.gov/genome/?term=txid9940[orgn], accessed on 3 August 2020) by Tophat2 tools [23,24]. Only the reads with a perfect match or one mismatch read were kept for further analysis. Differential expression gene (DEG) analysis was performed using the model based on the negative binomial distribution analysis of DESeq R package [25]. Genes with a *p*-value less than 0.05 were assigned as differentially expressed.

### 2.6. Data Analysis

The one-way analysis of variance was performed by using SPSS version 22.0 (SPSS, IBM, Inc., Chicago, IL, USA) to analyze the effect of feeding systems on FAs content. Significant differences among the GSF, GF, and SF groups were determined by Duncan’s post-hoc multiple comparisons test. Statistical significance was considered as *p* < 0.05.

Metabolome data were analyzed by XCMS program for peak alignment, retention-time correction, and peak-area extraction. The metabolite structure identification was performed through accurate mass matching (<25 ppm) and secondary spectrum matching methods. Metabolites with missing values exceeding 50% in the group and with extreme values were removed. The total peak area of the data was normalized to ensure the parallel comparison. The data were imported into software SIMCA-P 14.1 (Umetrics, Umea, Sweden), where it was subjected to multivariate data analysis, including unsupervised principal component analysis (PCA), discriminant analysis of squares (PLS-DA) with minimal supervision, and orthogonal partial least squares discriminant analysis (OPLS-DA). The variable importance in the projection (VIP) value of each variable in the OPLS-DA model was calculated to indicate its contribution to the classification. Metabolites with the VIP value >1 were further applied to Student’s *t*-test at univariate level to measure the significance of each metabolite, and a *p* value < 0.05 was deemed as statistically significant.

DEGs in each pair of the different feeding systems were functionally annotated by gene ontology (GO) analysis [26]. Physiological metabolism events and signal pathways of the DEGs were assessed using KOBAS software to test the statistical enrichments of the DEGs in Kyoto Encyclopedia of Genes and Genomes (KEGG) pathways [27]. GO and KEGG analysis results with corrected *p* value less than 0.05 were considered to be significantly different.

Correlation among the FA composition, differential metabolites, and differentially expressed genes was conducted by Pearson correlation analysis of SPSS version 22.0 (SPSS, IBM, Inc., Chicago, IL, USA). Significant correlation was considered as *p* < 0.05.

## 3. Results and Discussion

### 3.1. Fatty Acid Analysis

FAs with different chain lengths play different roles in animal products. Among them, volatile FAs with a chain length of 4–6 tend to be deposited in milk but less in meat. FAs with a chain length of 8–10, particularly branched-chain fatty acids, are the main flavor molecules in lamb [28]. Short-chain fatty acids (SCFAs) also can be responsible for cooked mutton flavor characteristics [29], but limited numbers of researchers have reported its potential roles in meat quality or flavor. Therefore, SCFA analysis in *LD* muscle may help to understand the meat characteristics of Tan lamb affected by different feeding systems.

In the present study, results indicate that no significant difference was observed in the content of short-chain fatty acid in Tan lambs among the GSF, GF, and SF groups (*p* > 0.05) (Table 2). However, the content of individual SCFAs showed the same trend in all three feeding systems, with the level of acetic acid > butyric acid > propionic acid > hexanoic acid > isobutyric acid > valeric acid > isovaleric acid. The presence of acetic acid, butyric acid, and propionic acid were found to be responsible for the changes in flavor and odor of milk and seafood [30,31], but limited studies have been performed on lamb meat. In this study, the proportion of acetic acid, butyric acid, and propionic acid to the total SCFA was 75.39%, 20.27%, and 1.38%, respectively. Moreover, some of the SCFA absorbed into the blood after rumen fermentation increased and exceeded the metabolic capacity of the liver. This would have accumulated and formed branched chain fatty acid deposited in the animal’s tissue [32]. Therefore, it is also possible that the SCFAs may serve as flavor compounds or precursors for the synthesis of other flavor fatty acids to affect meat quality.

FA content in muscle has been correlated with meat quality and human health [5,8,33]. Results of the effect of different feeding systems on the content of saturated fatty acids (SFA) are shown in Table 3 and indicated that the content of ∑SFA in both GSF and GF groups was significantly lower than that of SF group (*p* < 0.05), while no significant difference was found between GSF and GF groups. Several studies with similar feeding systems also have demonstrated that indoor feeding increased the SFA concentration [34,35,36]. The C16:0, as the major component of SFA, had a higher level in the stall-feeding lambs [17] which was harmful for human health [37]. For monounsaturated fatty acids (MUFA), lamb meat form the GSF group had lower ∑MUFA than that in the GF and SF group (*p* < 0.05). According to previous results, C18:1n9 (oleic acid) and C22:1n9 were higher in the meat from stall-feeding lamb when compared with graze-feeding system [17]. Oleic acid takes the highest percentage of MUFA, and the higher content of oleic acid might be associated with an increase of fat accumulation and the activity of stearoyl-CoA desaturase which is responsible for the synthesis of oleic acid from stearic acid in stall-feeding lambs [38].

Moreover, in the current study, the polyunsaturated fatty acids (PUFA) content were significantly higher in the GSF group than in the GF and SF groups (*p* < 0.05). These results are consistent with previous reports that pasture raised lambs had greater percentages of PUFA, especially n-3 FAs, with lower n-6/n-3 ratios apart from a greater percentage of conjugated linoleic acid (CLA) than lambs from closed feeding systems with concentrate and hay [39], which is favorable for human health. The linoleic acid and linolenic acid play important roles as PUFA precursors of odor-active compounds [40], these two FA increased in lambs from graze-feeding system, which may lead to less of the typical lamb flavor and fatty flavor in meat [17,41]. Additionally, compared to stall-feeding lambs, the lower ∑SFA/∑PUFA was found in lamb from both graze-feeding and time-limited graze-feeding groups (*p* < 0.05) which demonstrated an improvement of FA composition in these groups. Previous study reported similar result that there was a tendency for the ratio to be lower in graze-feeding system [42]. However, the mechanisms of FAs deposition in lamb meat from different feeding systems are not fully understood. Based on the factor of diet, a possible explanation is FA content in lamb meat has positive relationship with diet FA composition. The grazing lambs feed more forage and lower concentrate than those in the indoor feeding lambs, but the forage contained higher content of PUFA and lower content of MUFA and SFA than these in concentrate [35]. Collectively, feeding system is the main factor affecting the FA content, grazing Tan lambs provided meat with better PUFA, followed by time-limited feeding, and lastly stall feeding.

### 3.2. Bioinformatic Analysis of Metabolites

The metabolome of *LD* muscle was performed to focus on the changes of lipid-metabolism-related metabolites under different feeding systems. This analysis will provide a comprehensive overview of the metabolic profile to help understand the mechanisms of unsatisfactory FA content as a response to limited-grazing and stall-feeding systems. Principal component analysis (PCA) of the metabolites showed that the GSF, GF, and SF groups could be divided into three separate groups under positive-ion mode (Figure 1A). PCA analysis in the negative-ion mode indicated that the GF and GSF groups were overlapped, but both were significantly different from the SF group (Figure 1B). These results demonstrated that different feeding systems have an impact on the metabolites in the *LD* muscle of Tan lamb, and the species of metabolites produced by Tan lambs in the GF group are more similar to the GSF group than the SF group.

The number of differential metabolites between the GF and GSF groups in positive- and negative-ion modes was 268 and 102, respectively (Figure 2A,B). For the comparison of the SF and GSF groups, 443 and 419 differential metabolites were identified in positive and negative modes, respectively (Figure 2C,D). The results of the hierarchical cluster map of differential metabolites between the GF vs GSF and the SF vs GSF groups under positive- and negative-ion modes are shown in Figure 3A,B and Figure 3C,D, respectively. More specifically, compared to the GSF group, the lower concentration of metabolites such as arachidic acid, all cis-(6,9,12)-linolenic acid, oleic acid, and pentadecanoic acid were identified in both the GF and SF groups (*p* < 0.05). These results are consistent with the lower content of unsaturated fatty acid detected in the results of fatty acid content. Furthermore, the carnitine derivatives (acetylcarnitine, L-carnitine, and L-palmitoylcarnitine) are markers of lipid oxidation and involved in the process of fatty-acyl CoA transport into the mitochondria [43], which can enhance energy and physical function as well as play a critical role in lipid transfer and utilization [44]. In our study, a lower level of carnitine derivatives was observed in the GF and SF groups than in the GSF group, which suggested the depressed FA catabolism and the increased lipid accumulation in the meat from grazing limited and stall-feeding lambs [45].

Conversely, metabolites such as palmitic acid (C16:0), mevalonic acid, glycerol-3-phosphate (an essential precursor for glycerophospholipid and triglyceride synthesis [46,47]), and dihydroxyacetone phosphate (which can be converted into glycerol-3-phosphate and involved in lipid metabolism [48]) had higher concentrations in the groups of GF and SF than these in the group of GSF (*p* < 0.05). Therefore, the level of intermediate products for fat synthesis was higher in stall-feeding lambs than that in grazing lambs, which may have contributed to the increased level of SFA in stall-feeding lambs. Choline derivatives (phosphorylcholine, glycerophosphocholine, and cytidine 5’-diphosphocholine) play an important role in signal transduction as a source of lipid signaling molecules [49]. In the present study, choline derivative concentrations showed a higher level in both the GF and SF groups than in the GSF group (*p* < 0.05). It is widely known that choline or choline derivatives are required for the secretion of lipoprotein particles from the liver and intestine which decreases their lipid accumulation and concentration [50]. However, while there is limited information regarding the effect of choline or choline derivatives on muscle, it has been reported that choline deficiency increased the percentage of monounsaturated fatty acids at the expense of saturated fatty acids in muscle cells [51]. The lipid metabolism of different tissues and organs varies, and the function of choline and its derivatives on lipid metabolism and FA composition needs to be further studied.

To further explore the effect of feeding systems on the metabolic pathway in *LD* muscle of Tan lamb, KEGG analysis was performed on the differential metabolites (Figure 4A,B). The results indicated that most of the candidate metabolites with lower concentrations in the comparisons of both the GF and GSF groups and the SF and GSF groups were significantly enriched in the lipid-metabolism-related pathway—the biosynthesis of unsaturated fatty acids pathway (*p* < 0.05). In addition, compared to the GSF group, metabolites with lower concentration in the SF group were enriched in the biosynthesis of linoleic acid metabolism pathway (*p* < 0.05). Meanwhile, the majority of candidate differential metabolites with higher concentrations were enriched in the glycerophospholipid metabolism pathway (*p* < 0.05), which is closely related to lipid metabolisms [52]. The KEGG results also demonstrated that the FA metabolism of *LD* muscle was significantly regulated by different feeding systems in Tan lamb.

### 3.3. Bioinformatic Analysis of Transcriptome

Transcriptomic analysis was conducted in the present study to investigate the candidate genes involved in the lipid-metabolism-related processes under different feeding systems. PCA analysis indicated that the distributions of the GF and GSF groups were partially overlapped, while the SF group was non-overlapped with the GF and the GSF groups (Figure 5A). The results suggested the genes expression profile of *LD* muscle in Tan lamb was similar between the GF and GSF groups, but quite different in the SF group from that of the GSF and GF groups. These results are similar to the PCA analysis of the metabolites under different feeding systems in negative-ion mode. In addition, the results of hierarchical clustering and KEGG analysis of differential metabolites demonstrated similarity of the GF and GSF groups and the dissimilarity of the SF and GSF groups. Therefore, SF vs. GSF was chosen for the following transcriptome analysis.

A total of 788 differentially expressed genes (DEGs) were detected when comparing the SF with GSF groups, among these, 424 genes were significantly up-regulated and 364 genes were significantly down-regulated, respectively (Figure 5B) (*p* < 0.05). The heat map clustering results showed that samples from the same feeding system can appear in the same cluster (Figure 6). It further demonstrated that the gene expression profiles of the SF and GSF groups were obviously different. Gene expression modifications were observed in response to different production systems (fed-indoors vs. grazed on pasture) in skeletal muscle of steers [53]. Pasture types (such as Calafatal pasture vs. naturalized pasture and cultivated high-yielding pasture vs. semi-natural grassland) also demonstrated a significant influence on genes expression profile in lambs [54] and horses [55].

The GO functional annotation results revealed that it was possible to highlight some important candidate DEGs annotated to the biological processes involved in lipid metabolism and transport terms (Figure 7A). More specifically, DEGs such as *FABP3*, *PLA2R1*, and ERFE, were involved in the terms of positive regulation of fatty acid transport/regulation of fatty acid transport; and genes such as *FABP3*, *LIPC*, *LBP*, *APOC4*, *LDLR*, *SLC10A4*, *SLC10A6*, *DRD2*, and *LHB* were enriched in lipid transport pathways. *FABP3* (up-regulated in stall-feeding lambs) belongs to one of the intracellular fatty-acid-binding proteins, which was associated with lipid metabolism and fatty acid synthesis by acting as intracellular transport of hydrophobic intermediated and lipid metabolites through the membranes [56]. The *PLA2R1* (up-regulated) and *LIPC* (up-regulated) genes are involved in the delivery of lipids [57] and the metabolism of lipoproteins and triglycerides [58], respectively. The higher expression of the *LBP* gene could increase the content of myristic, palmitic acids and total SFA, and be responsible for fatty acid transportation between cells as a carrier or agent [56]. Furthermore, genes such as *APOC4*, *LDLR*, *SLC10A4*, and *SLC10A6*, act as a carriers of fatty acids, playing an important role in lipid metabolism [59,60,61]. The down-regulation of these genes in stall-feeding lambs suggested the depressed circulation of lipid levels, which thus may have improved the synthesis and accumulation of lipid in muscle. Based on the current data, the regulatory relationship between each gene and specific FA needs to be further investigated. In addition, the KEGG results demonstrated that the DEGs between the GSF and SF groups are enriched in lipid-associated pathways including the steroid biosynthesis and cholesterol metabolism pathways (Figure 7B). The steroid biosynthesis pathway is a key pathway related to lipid storage and metabolism, and may also be the key pathway in regulating differential lipid deposition [62]. The pathway was found significantly enriched by DEGs in muscle of cattle from different nutrition supplies [63]. In addition, the cholesterol metabolism pathway is also involved in lipid metabolism and affects FA synthesis [64,65]. These reports further confirmed that the DEGs between stall-feeding and grazing lambs observed in this study affect lipid metabolism, and they provide an insight for further study on the regulation of genes on FA content or lipid metabolism under different feeding systems.

### 3.4. Correlation Analysis

Results of Pearson correlation analysis among the FA contents, differential metabolites, and DEGs are illustrated in Figure 8. The concentrations of ∑FA, ∑MUFA, and ∑SFA showed significant negative correlation with metabolites such as cis-(6,9,12)-linolenic acid, arachidic acid, L-carnitine, and acetylcarnitine, and genes such as *LDLR* and *SLC10A6* (*p* < 0.05), but had significant positive correlation with the metabolites of glycerophosphocholine and glycerol-3-phosphate, and the *LIPC*, *ERFE*, *FABP3*, and *PLA2R1* genes (*p* < 0.05). These results suggest that lamb meat from stall-feeding system shows higher levels of ∑FA, ∑MUFA, and ∑SFA, which is consistent with previous research on differential metabolites and DEG functions. Moreover, there was significant negative correlation between the content of ∑PUFA and the metabolites of glycerophosphocholine and glycerol-3-phosphate, and the genes of *ERFE* and *FABP3*, but positive correlation between the content of ∑PUFA and the metabolites of cis-(6,9,12)-linolenic acid and acetylcarnitine (*p* < 0.05), which demonstrated that these metabolites and genes have an important effect on the regulation of ∑PUFA content in lamb meat. The results also indicated that some of the metabolites and genes play an opposite regulatory role between the content of ∑PUFA and the content of ∑FA, ∑MUFA, and ∑SFA, which will provide an insight on the control of FAs content by key metabolites and genes in future studies. In addition, the expression level of *LDLR* had a positive relationship with cis-(6,9,12)-linolenic-acid content (*p* < 0.05), which suggests that *LDLR* is a potential biomarker gene involved in the regulation of the cis-(6,9,12)-linolenic-acid content in Tan lamb meat.

## 4. Conclusions

In conclusion, grazing lambs presented a lower content of saturated fatty acids and monounsaturated fatty acids, but showed a higher content of polyunsaturated fatty acids and higher ∑PUFA/∑SFA. Moreover, differential metabolites under different feeding systems were mainly involved in lipid-metabolism-associated pathways including biosynthesis of unsaturated fatty acids and glycerophospholipid metabolism; the differentially expressed genes were enriched in the steroid biosynthesis and cholesterol metabolism pathways. Additionally, the differential metabolites and genes were associated with the content of ∑FA, ∑MUFA, ∑SFA, and ∑PUFA in lamb meat. These findings demonstrated that feeding system could regulate fatty acid composition through affecting lipid metabolism in meat of Tan lambs.

## Figures and Tables

**Figure 1 foods-10-02612-f001:**
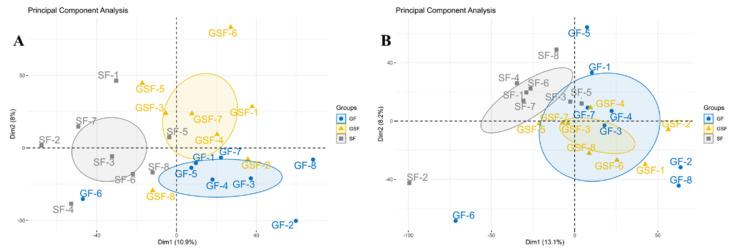
PCA analysis of metabolites under positive- (**A**) and negative- (**B**) ion modes in *LD* muscle of Tan lamb. GSF, graze-feeding group; GF, time-limited graze-feeding group; SF, stall-feeding group. *n* = 8 for each group.

**Figure 2 foods-10-02612-f002:**
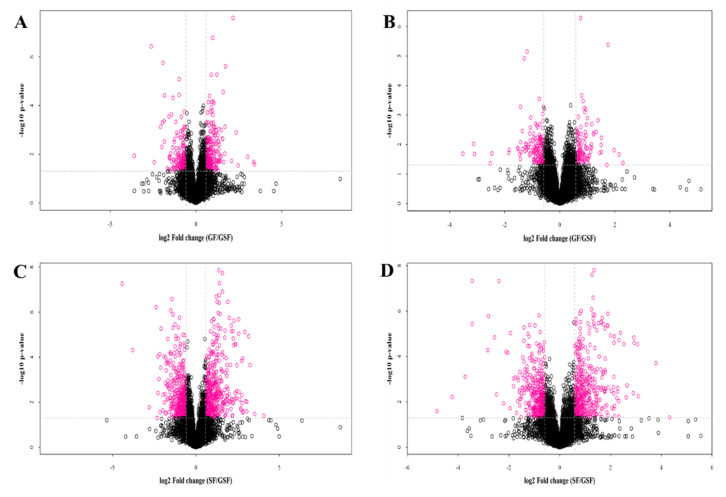
Volcano plot of Tan lamb metabolites under positive- and negative-ion modes in the GF vs GSF (**A**,**B**) and SF vs GSF (**C**,**D**) groups. GSF, graze-feeding group; GF, time-limited graze-feeding group; SF, stall-feeding group. *n* = 8 for each group. The black and purplish red points indicate non-significantly different metabolites and significantly different metabolites, respectively.

**Figure 3 foods-10-02612-f003:**
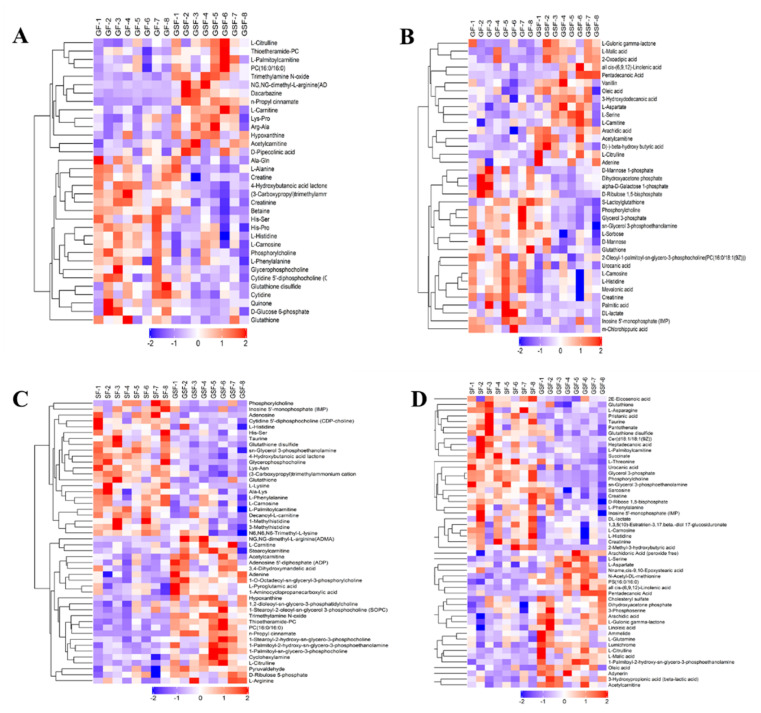
Hierarchical cluster results of differential metabolites under positive- and negative-ion modes in the GF vs GSF (**A**,**B**) and SF vs GSF (**C**,**D**) groups. GSF, graze-feeding group; GF, time-limited graze-feeding group; SF, stall-feeding group. *n* = 8 for each group.

**Figure 4 foods-10-02612-f004:**
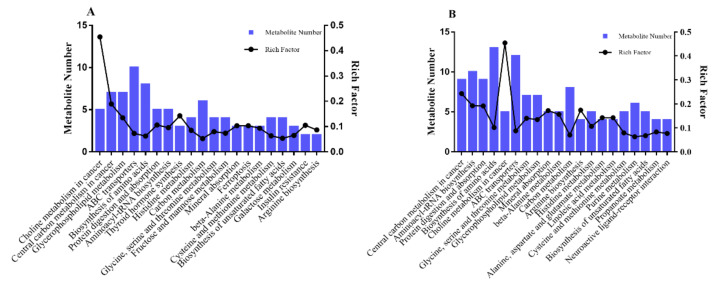
KEGG pathway enrichment analysis results of metabolites in the comparison of the GF and GSF groups (**A**) and the SF and GSF groups (**B**). *n* = 8 for each group.

**Figure 5 foods-10-02612-f005:**
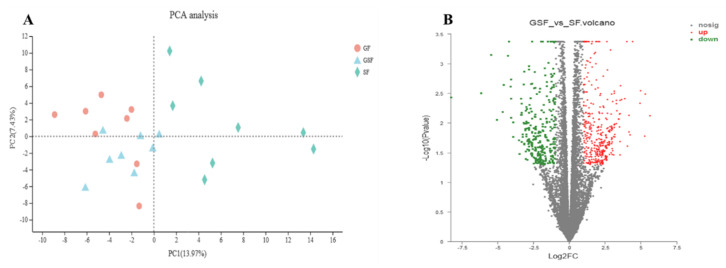
PCA analysis of samples from different feeding systems (**A**) and volcano map of gene expression profile in *LD* muscle of Tan lamb between the SF and GSF groups (**B**). The green, red, and blue points indicate significant down-regulated, up-regulated, and non-significant difference genes, respectively. GSF, graze-feeding group; SF, stall-feeding group; FC, fold change. *n* = 8 for each group.

**Figure 6 foods-10-02612-f006:**
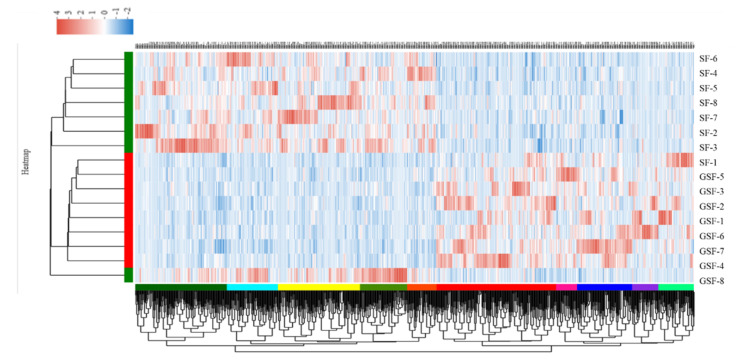
Cluster heat map in *LD* muscle of Tan lamb between the SF and GSF groups. GSF, graze-feeding group; SF, stall-feeding group. *n* = 8 for each group.

**Figure 7 foods-10-02612-f007:**
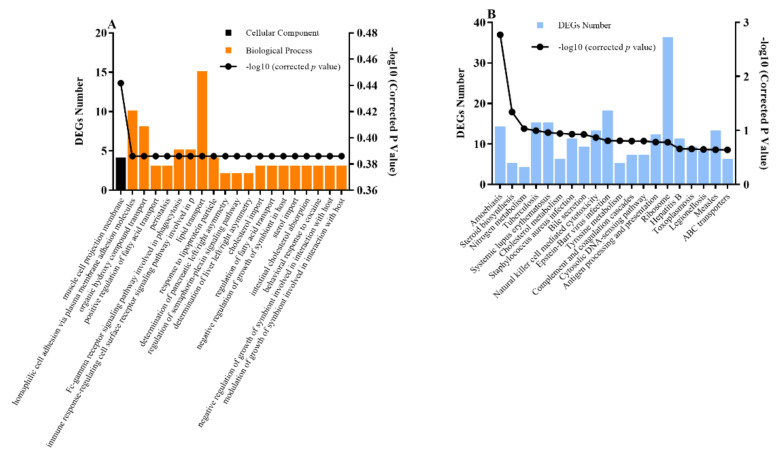
GO (**A**) and KEGG (**B**) analysis results of the DEGs between the SF and GSF groups. DEGs, differentially expressed genes. *n* = 8 for each group.

**Figure 8 foods-10-02612-f008:**
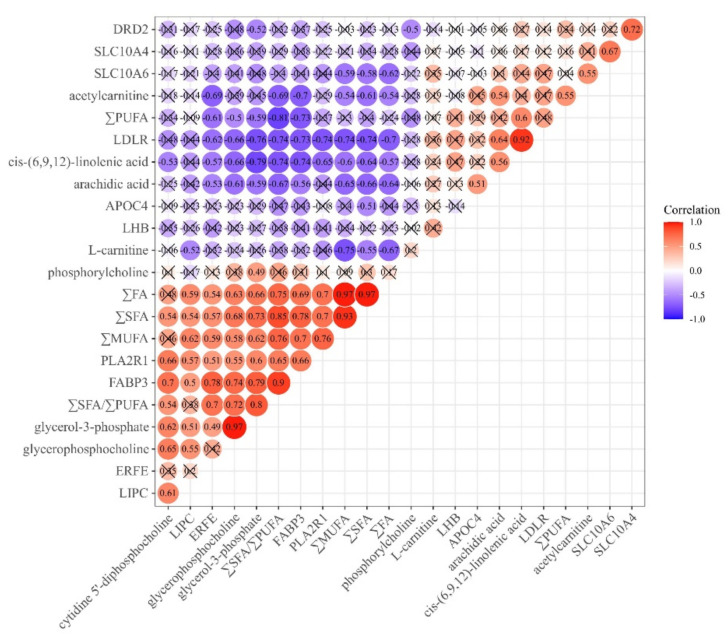
Pearson correlation analysis results for fatty acid composition, differential metabolites, and differentially expressed genes. Red and blue circles indicate significant positive- and negative-correlation, respectively. Circle with “×” indicates non-significant correlation.

**Table 1 foods-10-02612-t001:** Component of dietary formula and nutritional level (dry-matter basis).

Ingredients	Content (%)	Components	Content (%)
Pellets			
Corn	25.24	Dry matter	93.17
Alfalfa powder	25.00	Crude protein	13.96
Wheat bran	7.76	Crude fiber	22.74
Rapeseed cake	6.00	Ether extract	3.32
Soybean meal	8.00	Crude ash	9.55
NaHCO_3_	1.00	Calcium	0.60
Salt	1.00	Phosphorus	0.35
Premix ^(1)^	1.00	Metabolic energy (MJ/kg) ^(2)^	9.39
Hay			
Alfalfa hay	10.00		
Corn straw	15.00		
Total	100.00		

^(1)^ Note: Each 1 kg premix contained 320000 IU vitamin A, 10000 IU vitamin E, 300 mg copper, 5000 mg iron, 5000 mg zinc, 1000 mg manganese, 40 mg iodine, 10 mg cobalt, and 10 mg selenium. ^(2)^ The metabolic energy was calculated value.

**Table 2 foods-10-02612-t002:** Effect of feeding system on adipose tissue fatty acid content (μg/g) in the *LD* muscle of Tan lambs.

Item	GSF (*n* = 8)	GF (*n* = 8)	SF (*n* = 8)	SEM	*p* Value
Acetic acid (C2:0)	37.09	35.69	35.77	1.11	0.8570
Propionic acid (C3:0)	0.79	0.63	0.57	0.14	0.8093
Butyric acid (C4:0)	10.94	9.71	8.62	1.93	0.8954
Isobutyric acid (iso-C4:0)	0.52	0.44	0.41	0.03	0.3484
Valeric acid (C5:0)	0.23	0.18	0.22	0.02	0.6214
Isovaleric acid (iso-C5:0)	0.21	0.17	0.18	0.01	0.3305
Hexanoic acid (C6:0)	0.59	0.52	0.58	0.06	0.8885
Total SCFA	50.37	47.35	46.36	3.17	0.8760

GSF, graze-feeding group; GF, time-limited grazing with supplementary feeding group; SF, stall-feeding group. SCFA, short-chain fatty acids.

**Table 3 foods-10-02612-t003:** Effect of feeding system on muscle fatty acid content (mg/100 g of meat) and composition (proportion × 100) in the *LD* muscle of Tan lambs.

Item	GSF (*n* = 8)	GF (*n* = 8)	SF (*n* = 8)	SEM	*p* Value
∑SFA	1276.39 ^b^ (46.02)	1282.97 ^b^ (44.44)	1677.73 ^a^ (49.15)	49.871	<0.001
∑MUFA	1026.83 ^c^ (37.02)	1210.37 ^b^ (41.93)	1349.95 ^a^ (39.55)	37.488	<0.001
∑PUFA	469.28 ^a^ (16.92)	395.98 ^b^ (13.72)	384.72 ^b^ (11.27)	10.830	<0.001
∑SFA/∑PUFA	2.73 ^c^	3.25 ^b^	4.38 ^a^	0.156	<0.001
∑FA	2773.50 ^b^	2886.77 ^b^	3412.94 ^a^	73.841	<0.001

Different superscripts (a, b, and c) indicate significant differences on the same row. Data in parentheses indicates percentage of total FA. GSF, graze-feeding group. GF, time-limited graze-feeding group. SF, stall-feeding group. *n* = 8 for each group. SFA, saturated fatty acids; MUFA, monounsaturated fatty acids; PUFA, polyunsaturated fatty acids; FA, fatty acids.

## Data Availability

Data are not available in public datasets; please contact the authors.

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
