# Peer review of "Changes of Metabolites and Gene Expression under Different Feeding Systems Associated with Lipid Metabolism in Lamb Meat"

_foods, 2021, doi:10.3390/foods10112612_

Round 1

Reviewer 1 Report

The presented study is interesting trying to provide new information on the effect of feeding system on lipid metabolism in sheep meat

Extensive English language corrections are required.  It is difficult to understand what the authors are trying to report.

The authors should be very careful when they try to use other words because there is a complete change of the meaning. For example lines 45 and 46. Do you mean composition and not metabolism?

Introduction

It is quite long and not focused.

Lines 41-60: Please make it shorter. There is no need for an extensive report on the effect of diet on meat fatty acid composition and nutritional value.

There are differences between lamb and mutton (line 80)

Please write the aim of the study clearly

Materials and methods

Please provide information on the type of grass feeding

Please provide information on the LD muscle (rib no to rib no)

How did you prepare the homogenate (line 127). Did you use lean meat removing with the scalpel adipose tissue?

Please provide references for all used methods.

Results and discussion

Line 232: Did you perform the analyses on lean meat or adipose tissue? You are referring to adipose tissue

Line 284: optimum PUFA (how do you define optimum?  Do you mean desirable?)

Table 2: Please provide detailed fatty acid composition. You are referring to individual fatty acids throughout the text but there is no information on each fatty acid.

SFA/PUFA : you cannot use just one index to discuss the effect of diet on the nutritional value of meat. Even though, you need to discuss about this index and provide suitable references.

Figure 2: Please add information on which colour represents what.

My question is why, you did not compare the two "extremes" GF and SF

Please try to increase the size of each figure.

Conclusions

Please write it again because it is not supported by the aim of the study.  Especially, the concluding sentence. Was identifications of biomarkers the purpose of your study?

Author Response

Reviewer 1

The presented study is interesting trying to provide new information on the effect of feeding system on lipid metabolism in sheep meat

Extensive English language corrections are required. It is difficult to understand what the authors are trying to report.

Response to the reviewer: The language was edited by a native English speaker.

The authors should be very careful when they try to use other words because there is a complete change of the meaning. For example lines 45 and 46. Do you mean composition and not metabolism?

Response to the reviewer: Thanks for the reviewer’s comments, and we revised relevant sentences throughout the manuscript.

Introduction

It is quite long and not focused.

Response to the reviewer: The introduction was revised according to the reviewer’s suggestion.

Lines 41-60: Please make it shorter. There is no need for an extensive report on the effect of diet on meat fatty acid composition and nutritional value.

Response to the reviewer: This part was revised as suggested by the reviewer.

There are differences between lamb and mutton (line 80)

Response to the reviewer: “mutton” was replaced by “lamb meat’ in the revised manuscript.

Please write the aim of the study clearly

Response to the reviewer: The aim of the present study was revised.

Materials and methods

Please provide information on the type of grass feeding

Response to the reviewer: Information of the mainly grass species in the meadow was supplemented in the revised manuscript.

Please provide information on the LD muscle (rib no to rib no)

Response to the reviewer: The sentence was revised as “LD muscle was taken from the 8th to 12th ribs …”

How did you prepare the homogenate (line 127). Did you use lean meat removing with the scalpel adipose tissue?

Response to the reviewer: The information of homogenization process was supplemented in the revised manuscript. Adipose tissue on the surface of LD muscle was removed before sampling. Intermuscular fat was extract for fatty acid determination.

Please provide references for all used methods.

Response to the reviewer: References were added as suggested by the reviewer.

Results and discussion

Line 232: Did you perform the analyses on lean meat or adipose tissue? You are referring to adipose tissue

Response to the reviewer: The analyses were performed on LD muscle, and the sentence was revised.

Line 284: optimum PUFA (how do you define optimum? Do you mean desirable?)

Response to the reviewer: The sentence was corrected as “…provided meat with better PUFA”.

Table 2: Please provide detailed fatty acid composition. You are referring to individual fatty acids throughout the text but there is no information on each fatty acid.

Response to the reviewer: The data of individual fatty acids was published, so we provided the total content in the present study, and cited and discussed the important results of individual fatty acids from the published paper in the present manuscript. In addition, the result of individual fatty acids was not the main aim of the present study, and most of the expression “fatty acid composition” were replaced by “fatty acid content”. (Zhao, X.G.; Guo, Y.P.; Liu, M.; Zhang, C.; Luo, H.L. Effect of desert steppe grazing on slaughtering performance and meat quality of Tan sheep. Pratacultural Science (in Chinese), 2021, 38, 554–561.)

SFA/PUFA: you cannot use just one index to discuss the effect of diet on the nutritional value of meat. Even though, you need to discuss about this index and provide suitable references.

Response to the reviewer: Thanks for the reviewer’s comment. The nutritional value of meat was discussed by using the results in table 3 in the revised manuscript.

Figure 2: Please add information on which colour represents what.

Response to the reviewer: The information was supplemented as suggested by the reviewer.

My question is why, you did not compare the two "extremes" GF and SF

Response to the reviewer: Thanks for your comments. Two reasons were considered in the present study: firstly, based on the results of fatty acid, the “extremes” difference indicated in the GSF and SF groups; secondly, the PCA analysis results of transcriptome and metabolomics showed the sample variation in the GF group was larger than the others.

Please try to increase the size of each figure.

Response to the reviewer: Thanks for the reviewer’s suggestion. The size of the figures was affected by the format of the manuscript. The figures can be read clearly by zooming in.

Conclusions

Please write it again because it is not supported by the aim of the study. Especially, the concluding sentence. Was identifications of biomarkers the purpose of your study?

Response to the reviewer: The conclusion was reorganized as suggested by the reviewer.

Reviewer 2 Report

It's a bit confusing the various nominations for lipids. Sometimes it's the fatty acid profile, sometimes its metabolomic, and sometimes it's transcriptomic. Please try to use homogeneousterms to refer to specific analyzes. In the discussion it is easy to differentiate these elements.

Abstract:

Add P-values in your results.

Line (17): Add “sex” of the animals

Line (18): Add “age” of the animals

Line (30): What is KEGG? Describe the full name here.

Line (30-32): With which feeding system?

Line (34-37): The conclusion must be specific. I recommend describing which feeding system was better.

Introduction

The first and second paragraphs are repetitive and contain the same information, although written in different ways.

The introduction seems like a disorganized collection of information. Rewrite the introduction following a line of reasoning. Also, rewrite the objective of the study making it clear. Because it seems as if they don't know how to describe the correct objective of the manuscript despite knowing what the general idea of the manuscript is.

Material and methods:

Line (118): Describe the method used for slaughter

Line (127): homogenate meat? Fat? Ruminal content?

Line (128): evenly? Was the isopropyl a solid? Or a liquid?

Line (141): Improve fatty acid measure methodology. Sample weight, reagents volume, standards use, etc. Follow the methodology of short fatty acids

Line (181): Use a citation for the conventional method

Table 1: Correct column headings. Ingredients; Content (%); Component or nutrient; Content (%)

Results and discussion

The discussion largely resembles a review of the literature. The discussion is a part of the manuscript where the results found are biologically described.

The first paragraphs reveal the results of the tables and indicate that the results are affected by the feeding system. But they will not explain how the production system promoted these results. E.g.: Why does the stall system promote higher SFA deposition?

Line (232): What studies?

Conclusion:

The conclusion is a text that belongs to the topic of results and not to the conclusion. Rewrite the conclusion being more specific.

Author Response

Reviewer 2

It's a bit confusing the various nominations for lipids. Sometimes it's the fatty acid profile, sometimes it’s metabolomic, and sometimes it's transcriptomic. Please try to use homogeneousterms to refer to specific analyzes. In the discussion it is easy to differentiate these elements.

Response to the reviewer: Fatty acids are an important component of muscle lipids. In the present study, the fatty acid content was measured, and the metabolomics and transcriptomic analysis were used to study the metabolites and genes which involved in the lipid metabolism, and their relationship with the fatty acid content. We corrected the manuscript to make it clearer.

Abstract:

Add P-values in your results.

Response to the reviewer: P- values were added in the revised manuscript.

Line (17): Add “sex” of the animals

Response to the reviewer: The information of “male lambs” was added in the revised manuscript.

Line (18): Add “age” of the animals

Response to the reviewer: The information of “4-month-old” was added in the revised manuscript.

Line (30): What is KEGG? Describe the full name here.

Response to the reviewer: The full name of KEGG is Kyoto Encyclopedia of Genes and Genomes. Information was supplemented.

Line (30-32): With which feeding system?

Response to the reviewer: The DEGs were found by the comparison of the GSF group with the SF group. The sentence was corrected.

Line (34-37): The conclusion must be specific. I recommend describing which feeding system was better.

Response to the reviewer: The conclusion was revised according to the reviewer’s suggestion.

Introduction

The first and second paragraphs are repetitive and contain the same information, although written in different ways.

Response to the reviewer: The introduction was revised as suggested by the reviewer.

The introduction seems like a disorganized collection of information. Rewrite the introduction following a line of reasoning. Also, rewrite the objective of the study making it clear. Because it seems as if they don't know how to describe the correct objective of the manuscript despite knowing what the general idea of the manuscript is.

Response to the reviewer: The introduction was revised to make it clearer.

Material and methods:

Line (118): Describe the method used for slaughter

Response to the reviewer: The slaughter method was supplemented in the revised manuscript.

Line (127): homogenate meat? Fat? Ruminal content?

Response to the reviewer: The sentence was corrected as “Approximately 100 mg of LD muscle sample (adipose tissue on the surface of LD muscle were removed) was thawed at room temperature and homogenized with 2 ml pre-cooled isopropyl ether.”

Line (128): evenly? Was the isopropyl a solid? Or a liquid?

Response to the reviewer: The isopropyl is liquid. The sentence was revised as the previous answer.

Line (141): Improve fatty acid measure methodology. Sample weight, reagents volume, standards use, etc. Follow the methodology of short fatty acids

Response to the reviewer: Thanks for the reviewer’s suggestion, we improved the description of fatty acid measurement, and added references.

Line (181): Use a citation for the conventional method

Response to the reviewer: The information of RNA extraction was corrected.

Table 1: Correct column headings. Ingredients; Content (%); Component or nutrient; Content (%)

Response to the reviewer: The headings was corrected according to the reviewer’s comment.

Results and discussion

The discussion largely resembles a review of the literature. The discussion is a part of the manuscript where the results found are biologically described.

Response to the reviewer: We corrected the results and discussion part.

The first paragraphs reveal the results of the tables and indicate that the results are affected by the feeding system. But they will not explain how the production system promoted these results. E.g.: Why does the stall system promote higher SFA deposition?

Response to the reviewer: The results caused by feeding systems mainly due to the amount of grass and concentrate intake. More information was provided in the revised manuscript.

Line (232): What studies?

Response to the reviewer: The sentence was revised.

Conclusion:

The conclusion is a text that belongs to the topic of results and not to the conclusion. Rewrite the conclusion being more specific.

Response to the reviewer: The conclusion part was revised according to the reviewer’s suggestion.

Reviewer 3 Report

Although I am not a native English speaker the manuscript needs be grammar revised.

Lines 47 – 60 These sentences are out of the context once are based in studies in beef and pork. This part of the introduction should be revised

Lines 65 – 68 This sentence is completely out of the context. Remove it, please.

Lines 68 – 96 – It is difficult to follow the text and what the authors intend to study in terms of the state of the art of the subject and the formulation of the study objective. The introduction must be revised, synthesized, and rationalized.

Line 108: What does nutrition level % mean? Explain the concept, please.

The final body weight should be presented as well as the average daily gain in each feeding system

Also the carcass weight, the ultimate carcass pH and cielab color attributes (particularly the L* and Chroma) should be provided for each feeding system.

126 – 149 The method of fat extraction and fatty acid methylation as well as the gas chromatography conditions should be widely and carefully described. Why the short chain fatty acids analyzed were separate from the other fatty acids? Did the chromatography analyze conditions the same? Anyway, the chromatography conditions should be described

204 – 229: Statistical analysis must be clarified. There are several methodologies without any specific proposal. First, a one-way analysis of variance without specifying the model and the analyzed effect. A PCA, a PLS and a discriminant analysis with what objective? After an One-dimensional statistical analysis including Student’s t-test and multiple of variation analysis!

Results:

The fatty acids in Table 2 must follow the delta nomenclature even though the common names could be referred in the text. There is no table of non-short chain fatty acids, and this data must be provided. The data should be expressed in percentage of total fat extracted.

The lipid quality analysis performed does not follow the normal indexes presented in the bibliography (n-6 / n-3; PUFA / SFA; the IA and IT indexes). And the discussion of the results must be carried out with other studies on these assumptions.

The conclusion on fatty acid composition should be rewritten after these alterations.

Author Response

Reviewer 3

Although I am not a native English speaker the manuscript needs be grammar revised.

Response to the reviewer: The language was edited by a native English speaker.

Lines 47 – 60 These sentences are out of the context once are based in studies in beef and pork. This part of the introduction should be revised

Response to the reviewer: The introduction was reorganized.

Lines 65 – 68 This sentence is completely out of the context. Remove it, please.

Response to the reviewer: The sentence was removed according to reviewer’s suggestion.

Lines 68 – 96 – It is difficult to follow the text and what the authors intend to study in terms of the state of the art of the subject and the formulation of the study objective. The introduction must be revised, synthesized, and rationalized.

Response to the reviewer: The introduction was revised to make it clearer.

Line 108: What does nutrition level % mean? Explain the concept, please.

Response to the reviewer: The words “nutrition level %” was replaced by “content %” according to another reviewer’s suggestion. The “nutrition level %” means the percentage of nutrients in diet. E.g.: The content of crude protein is 13.96% means the 100 g diet contained 13.96 g crude protein.

The final body weight should be presented as well as the average daily gain in each feeding system

Also the carcass weight, the ultimate carcass pH and cielab color attributes (particularly the L* and Chroma) should be provided for each feeding system.

Response to the reviewer: Thanks for the reviewer’s comments. Data of final body weight and carcass weight was published, and the results was cited in the “2.1. Experimental Design and Sample Collection” part. Data of carcass pH and color attributes was published in a Chinese journal (Zhao, X.G.; Guo, Y.P.; Liu, M.; Zhang, C.; Luo, H.L. Effect of desert steppe grazing on slaughtering performance and meat quality of Tan sheep. Pratacultural Science (in Chinese), 2021, 38, 554–561.), and no significant difference was found among the three feeding systems. In addition, we believe that the results of pH and color are not closely related to the aim of the present study.

126 – 149 The method of fat extraction and fatty acid methylation as well as the gas chromatography conditions should be widely and carefully described. Why the short chain fatty acids analyzed were separate from the other fatty acids? Did the chromatography analyze conditions the same? Anyway, the chromatography conditions should be described

Response to the reviewer: Thanks for the reviewer’s suggestion, we improved the description of fatty acid measurement. This part was referenced the previous publication in our team, so we gave a brief description, and added related references. The chromatography analyze conditions between short chain fatty acids and other fatty acids were different, such as the running program and internal standard.

204 – 229: Statistical analysis must be clarified. There are several methodologies without any specific proposal. First, a one-way analysis of variance without specifying the model and the analyzed effect. A PCA, a PLS and a discriminant analysis with what objective? After an One-dimensional statistical analysis including Student’s t-test and multiple of variation analysis!

Response to the reviewer: The description of statistical analysis was improved according to the reviewer’s comments.

Results:

The fatty acids in Table 2 must follow the delta nomenclature even though the common names could be referred in the text. There is no table of non-short chain fatty acids, and this data must be provided. The data should be expressed in percentage of total fat extracted.

Response to the reviewer: The delta nomenclature of fatty acids in Table 2 was supplemented according to the reviewer’s suggestion. The data of individual fatty acids was published, so we provided the total content in the present study, cited and discussed the important results of individual fatty acids from the published paper in the present manuscript. (Zhao, X.G.; Guo, Y.P.; Liu, M.; Zhang, C.; Luo, H.L. Effect of desert steppe grazing on slaughtering performance and meat quality of Tan sheep. Pratacultural Science (in Chinese), 2021, 38, 554–561.). In addition, we believed that the result of individual fatty acids was not the main aim of the present study.

For the question about the expression of fat acids content, we believe that FA content expressed as the unit of μg/g or mg/100g of fresh meat could reflect the real content of FA in meat (In recent years, journal such as “Meat Science” recommends this expression). The data expressed as percentage reflects the fatty acid take the percentage of total FA, but not the meat. Moreover, we supplemented the results of FA which expressed as the percentage of total FA in the revised manuscript as suggested by the reviewer.

The lipid quality analysis performed does not follow the normal indexes presented in the bibliography (n-6 / n-3; PUFA / SFA; the IA and IT indexes). And the discussion of the results must be carried out with other studies on these assumptions.

Response to the reviewer: As previous explanation, we listed the indexes not published before, and discussed it in the present study. The discussion part also improved according to the reviewer’s comments.

The conclusion on fatty acid composition should be rewritten after these alterations.

Response to the reviewer: The conclusion part was revised according to the reviewer’s suggestion.

Round 2

Reviewer 1 Report

English language corrections are still required

Line 44: are (present tense)

Lines 73-74: Please add the names of the fatty acids

Lines 78-79: Therefore, 76 the objective of this study was to investigate the effect of different feeding systems on FA 77 content, metabolites and gene expression and their relationship in the LD muscle of Tan lambs.

Please delete text in bold letters.

Line 115: Measurement of Short Chain Fatty Acids Please change to

Analysis of Short Chain Fatty Acid composition

Please provide the reference of the method and provide information on methyester preparation

Line 131: Fatty Acids Measurement Please change to

Analysis of muscle fatty acid composition

Please provide the method reference.

Line 134: identification.  This is wrong. Chloroform methanol are used for fat extraction.  Please correct the method and provide the steps i.e. preparation of methylesters, etc.

Please provide a table of individual fatty acid content.  You only refer to the different classes of fatty acids (Table 1) but throughout the text you refer to individual fatty acids.

Table 2: Please change title as follows;

Effect of feeding system on adipose tissue fatty acid content (μg/g) in the LD muscle of Tan lambs

Table 3: Please change title as follows;

Effect of feeding system on muscle fatty acid content (mg/ 100g of meat) and composition (proportion X 100) in the LD muscle of Tan lambs

Author Response

Dear Reviewer,

Thank you for helping us improve the quality of manuscript (ID: foods-1417537). We gratefully appreciate the constructive comments made by you. We revised the manuscript according to your comments and suggestions. Any changes of the manuscript were indicated in red in the revised manuscript and listed as follows.

English language corrections are still required

Response to the reviewer: Thank you for your comments. We have checked the language of the manuscript again.

Line 44: are (present tense)

Response to the reviewer: The word was revised according to the reviewer’s suggestion.

Lines 73-74: Please add the names of the fatty acids

Response to the reviewer: The names of the fatty acids were supplemented as suggested by the reviewer.

Lines 78-79: Therefore, 76 the objective of this study was to investigate the effect of different feeding systems on FA 77 content, metabolites and gene expression and their relationship in the LD muscle of Tan lambs. Please delete text in bold letters.

Response to the reviewer: The description of “the LD muscle of” was deleted.

Line 115: Measurement of Short Chain Fatty Acids Please change to Analysis of Short Chain Fatty Acid composition

Please provide the reference of the method and provide information on methyester preparation

Response to the reviewer: The title was revised according to the reviewer’s suggestion. Reference of the method was added in the revised manuscript. More information about the preparation of short chain fatty acids determination was provided.

Line 131: Fatty Acids Measurement Please change to Analysis of muscle fatty acid composition

Please provide the method reference.

Response to the reviewer: The title was revised according to the reviewer’s suggestion. References involved in the methods were supplemented.

Line 134: identification. This is wrong. Chloroform methanol are used for fat extraction. Please correct the method and provide the steps i.e. preparation of methylesters, etc.

Response to the reviewer: Thanks for the reviewer’s suggestion. The sentence was corrected according to the reviewer’s suggestion. The preparation of fatty acid methyl esters was supplemented in the revised manuscript.

Please provide a table of individual fatty acid content. You only refer to the different classes of fatty acids (Table 1) but throughout the text you refer to individual fatty acids.

Response to the reviewer: Thanks for the reviewer’s comments of this question. The data of individual fatty acids has been published (part of the results were cited in the present study), so it can’t present in this manuscript. However, we provided the total content of different categories of FA in table 3 and discussed their relationships with the differential metabolites and differentially expressed genes, which is consistent with the objective of this study. In addition, in the previous revised manuscript, we mainly discussed the results about total content of SFA, MUFA and PUFA, not the individual fatty acid content.

Table 2: Please change title as follows;

Effect of feeding system on adipose tissue fatty acid content (μg/g) in the LD muscle of Tan lambs

Response to the reviewer: The title of table 2 was revised according to the reviewer’s suggestion.

Table 3: Please change title as follows;

Effect of feeding system on muscle fatty acid content (mg/ 100g of meat) and composition (proportion X 100) in the LD muscle of Tan lambs

Response to the reviewer: The title of table 3 was revised according to the reviewer’s suggestion.

Reviewer 2 Report

The manuscript has been improved. The authors did a good job and I am satisfied with the results.

Author Response

The manuscript has been improved. The authors did a good job and I am satisfied with the results.

Response to the reviewer: Thank you for your comments.

Reviewer 3 Report

Authors answered the questions even though I think the quality of the manuscript could be improved.

Author Response

Authors answered the questions even though I think the quality of the manuscript could be improved.

Response to the reviewer: We gratefully appreciate your constructive comments and suggestions, we strive to improve the quality of the manuscript. Thanks again!